# Advancing equitable access to digital mental health in the Asia-Pacific region in the context of the COVID-19 pandemic and beyond: A modified Delphi consensus study

Jill K. Murphy[1]*, Shirley Saker[2], Promit Ananyo Chakraborty[3], Yuen Mei (Michelle) Chan[4], Erin E. Michalak[1], Matias Irrarazaval[5], Mellissa Withers[2], Chee H. Ng[6], Amna Khan[1], Andrew Greenshaw[7], John O'Neil[8], Vu Cong Nguyen[9], Harry Minas[10], Arun Ravindran[11], Angela Paric[11], Jun Chen[12], Xing Wang[12], Tae-Yeon Hwang[13], Nurashikin Ibrahim[14], Simon Hatcher[15], Vanessa Evans[1], Raymond W. Lam[1]

1 Department of Psychiatry, University of British Columbia, Vancouver, British Columbia, Canada, 2 Department of Population and Public Health Sciences, University of Southern California, Los Angeles, California, United States of America, 3 School of Population and Public Health, University of British Columbia, Vancouver, British Columbia, Canada, 4 School of Nursing, The University of Hong Kong, Hong Kong Special Administrative Region, China, 5 Pan American Health Organization, Santiago, Chile, 6 Department of Psychiatry, University of Melbourne, Melbourne, Australia, 7 Department of Psychiatry, University of Alberta, Edmonton, Alberta, Canada, 8 Faculty of Health Sciences, Simon Fraser University, Burnaby, British Columbia, 9 Institute of Population, Health and Development, Hanoi, Vietnam, 10 Centre for Mental Health, University of Melbourne, Melbourne, Australia, 11 Centre for Addiction and Mental Health/ Department of Psychiatry, University of Toronto, Toronto, Ontario, Canada, 12 Shanghai Mental Health Centre, Shanghai, China, 13 Korea Foundation for Suicide Prevention, Seoul, South Korea, 14 Mental Health, Injury and Violence Prevention and Substance Abuse Sector Section, Ministry of Health, Kuala Lumpur, Malaysia, 15 Department of Psychiatry, University of Ottawa, Ottawa, Ontario, Canada

* jill.murphy@ubc.ca

**Data Availability Statement:** The survey data are publicly available on Open Science Framework

## Abstract

The COVID-19 pandemic had an unprecedented impact on global mental health and well-being, including across the Asia-Pacific. Efforts to mitigate virus spread led to far-reaching disruption in the delivery of health and social services. In response, there was a rapid shift to the use of digital mental health (DMH) approaches. Though these technologies helped to improve access to care for many, there was also substantial risk of access barriers leading to increased inequities in access to mental health care, particularly among at-risk and equity-deserving populations. The objective of this study was to conduct a needs assessment and identify priorities related to equitable DMH access among at-risk and equity-deserving populations in the Asia Pacific region during the first year of the COVID-19 pandemic. The study consisted of a modified Delphi consensus methodology including two rounds of online surveys and online consultations with stakeholders from across the region. Study participants included policy makers, clinicians and service providers, and people with lived experience of mental health conditions. Results demonstrate that vulnerabilities to negative mental health impacts and access barriers were compounded during the pandemic. Access barriers included a lack of linguistically and culturally appropriate DMH options, low mental health literacy and poor access to technological infrastructure and devices, low levels of awareness and trust of DMH options, and lack of policies and

(DOI 10.17605/OSF.IO/KWH52). To protect participant confidentiality, including due to concerns about the inclusion of identifiers and sensitive information in a relatively small dataset, we have not made the transcripts from the qualitative consultations publicly available.

**Funding:** This study was funded by a COVID-19 Knowledge Synthesis Grant from the Canadian Institutes of Health Research (FRN: 171735). JKM and RWL were co-principal investigators; MI, MW, CHN, AG, JON, VCN, HM, AR, AP, JC, TYH, NI, SH were all co-investigators. PAC and AK received salary support as a Research Assistant from this grant. The APEC Digital Hub for Mental Health has received core funding from the Public Health Agency of Canada's International Health Grants Program. JKM, RWL, EEM, AG, CHN, AR are members of the APEC Digital Hub Executive. VE has received salary support as a Research Manager for the APEC Digital Hub for Mental Health. The funders had no role in study design, data collection and analysis, decision to publish, or preparation of the manuscript.

**Competing interests:** "I have read the journal's policy and the authors of this manuscript have the following competing interests: EEM has received funding from Otsuka-Lundbeck for patient educational activities. CHN had served as a consultant for Janssen-Cilag, Lundbeck, Grunbiotics, Servier and Eli Lilly, received research grants from National Health and Medical Research Council and Medical Research Future Funds, and speaker honoraria from Servier, Lundbeck, Eli Lilly, Janssen-Cilag, Astra-Zeneca, and Pfizer – all unrelated to this study. RWL has received honoraria for ad hoc speaking or advising/consulting, or received research funds, from: Abbvie, Asia-Pacific Economic Cooperation, Bausch, BC Leading Edge Foundation, Brain Canada, Canadian Institutes of Health Research, Canadian Network for Mood and Anxiety Treatments, CAN-BIND Solutions, Carnot, Grand Challenges Canada, Healthy Minds Canada, Janssen, Lundbeck, Medscape, Michael Smith Foundation for Health Research, MITACS, Neurotorium, Ontario Brain Institute, Otsuka, Pfizer/Viatris, Shanghai Mental Health Center, Sunnybrook Health Sciences Centre, Unity Health, Vancouver Coastal Health Research Institute, and VGH-UBCH Foundation. This does not alter our adherence to PLOS Global Public Health policies on sharing data and materials JKM, SS, YMC, PAC, MI, MW, AK, AG, JON, VCN, HM, AR, AP, JC, XW, TYH, NI, SH, and VE have declared that no competing interests exist.

guidelines to support effective and equitable delivery of DMH. Recommendations to improve equitable access include ensuring that diverse people with lived experience are engaged in research, co-design and policy development, the development and implementation of evidence-based and equity-informed guidelines and frameworks, clear communication about DMH evidence and availability, and the integration of DMH into broader health systems. Study results can inform the development and implementation of equitable DMH as its use becomes more widespread across health systems.

## Introduction and background

The novel coronavirus pandemic (COVID-19) disrupted the lives of hundreds of millions of people and affected ways of socializing, working, and living globally, including across the Asia-Pacific region [1,2]. The pandemic exacerbated existing mental health conditions and resulted in increased incidence of mental illness and psychological distress [3]. A 2021 meta-analysis [2] which included 66 English and Chinese language studies from early in the pandemic found that the pooled prevalence of depression was 31.4%, and the prevalence of anxiety, distress, and insomnia was 31.9%, 41.1% and 37.9% respectively, all higher than the average rates prior to the pandemic. A systematic review examining the reported global prevalence of depression and anxiety disorders in 2020–2021 [4] found that the pandemic led to an increase of 27.6% in cases of major depressive disorder and of 26.7% in anxiety disorder. The substantial mental health impact of the pandemic has occurred in the context of already strained and under-resourced mental health systems. With the exception of New Zealand, the number of psychiatrists across the region is lower than the Organization for Economic Cooperation and Development (OECD) average of 18.1 per 100,000 population and there is limited integration of mental health care into community and primary care settings [5,6]. The gap in mental health care access is particularly challenging in low-and-middle income countries [7].

Many factors have contributed to the mental health impact of the pandemic. Worries related to being infected with COVID-19, experiences of illness and grief, increasing work pressure, lifestyle changes and worsening socioeconomic conditions all acted as risk factors for poor mental health [8–10]. Although precautions like social distancing and stay at home orders were essential measures taken by governments to contain the virus, they also led to challenges including isolation and increased feelings of loneliness which contributed to negative mental health impacts, including an increase in the prevalence of mental health conditions like depression [3,11].

Though the pandemic has had a widespread impact on mental health, some populations, particularly equity-deserving groups who faced social and systemic marginalization prior to the pandemic experienced an elevated risk of negative mental health impacts. In the United States, for example, racial and ethnic minorities experienced mental health risk factors including elevated risk of COVID-19 infection and mortality [12–14], increased job loss and related economic stressors, higher representation among frontline workers, and increased racialized violence and discrimination [15,16]. These factors are compounded by the effects of persistent inequities resulting from social and structural determinants of health including systemic racism, historic traumas, stigma, and high costs of accessing health care [17,18]. Healthcare workers also experienced higher rates of mental distress. For example, a study conducted in Indonesia in 2020 found that 83% of healthcare workers suffered from moderate-severe burnout syndrome during the pandemic, and approximately 40% had moderate-severe loss of

empathy, which was twice the pre-pandemic rate [19]. Individuals with pre-existing mental, neurological and substance use conditions were also at higher risk, including due to the partial or complete disruption of mental health services during the pandemic [20]. Another at risk population was youth and children. A study conducted in Canada in 2020 found that adolescent substance use increased during the pandemic and was correlated with COVID-19 fears and depressive symptoms [21]. Studies from the US [22], China [23,24] and Canada [21] conducted early in the pandemic identified mental health risk factors such as separation from social networks and school closures, as well as increases in negative lifestyle factors among youth including increased screen time [24], decreased exercise [25] and increased substance use [21]. A study conducted in China in 2020 showed that depression rates among youth increased from 13.2% (pre-pandemic) to 22.28% [26]. Finally, older adults were also at increased risk of having negative mental health outcomes. For example, in Australia, COVID-19 physical restrictions limited psychosocial support for seniors with the cessation of community services and communal activities, contributing to this mental health risk [27].

With face-to-face health and social services brought to a halt and the increasing mental health need among populations, there was a rapid shift to using digital technologies to provide mental health care early in the pandemic [1,28]. Digital health encompasses a variety of technologies including telephone, SMS (text message), mobile applications (apps), wearables (such as smart watches), and online tools. Digital mental health (DMH) leverages these technologies to support mental health including through screening, promotion, prevention, treatment, and support. Prior to the pandemic, though evidence supported the effectiveness of many DMH interventions, uptake among health systems, clinicians and patients was low [29]. Despite the opportunities presented by the pandemic to advance evidence based DMH services, the rapid shift to DMH application raised concerns about equitable access to these technologies by many populations in the Asia-Pacific region. Although comprehensive data on DMH access across the APEC region is unavailable, evidence suggests that while digital technology use in the region is widespread, access to digital technologies in general is not equal among many sub-populations. For example, the Canadian Radio-television and Telecommunications Commission (CRTC) reports that while broadband internet is accessible to 91.4% of the general Canadian population, only 62% of Canadians in rural communities and 43.3% of First Nations have adequate broadband access.[30] A study from China also found significantly lower Internet access among people in rural areas compared with people living in urban centers [31]. In Indonesia, 80% of people who are not able to access the Internet live in rural areas [32]. People who experience marginalization are also at risk of facing barriers to DMH access [1,33,34]. Numerous factors may act as access barriers to accessing DMH technologies, such as poverty, homelessness, insufficient health system resources and the lack of provider training and cultural competency to interact with diverse patients and communities [1,34]. DMH access, therefore, is interconnected with social determinants of health, along with the social, cultural, and economic realities which influence health equity [34,35]. Though pandemic precautions have eased, DMH options remain pervasive and offer a promising opportunity to increase access to mental health care across the Asia-Pacific region and worldwide. Equitable access, however, must be prioritized in the design, delivery, and governance of DMH.

Given the increased burden of mental health conditions, the rapid increase in the use of DMH and pervasive mental health system capacity challenges, there is a need to ensure that DMH options are accessible to all. Evidence regarding equitable access to DMH, however, is relatively limited [36]. The objective of this study was to conduct a needs assessment and identify priorities related to mental health among at-risk and equity-deserving populations in the Asia-Pacific region during the first year of the COVID-19 pandemic with a focus on Asia-Pacific Economic Cooperation (APEC) member economies (APEC uses the term 'economies'

instead of 'countries'). This study was undertaken by members of the APEC Digital Hub for Mental Health's ('the Digital Hub', https://mentalhealth.apec.org/) Disaster Resilience and Trauma working group and focuses on the APEC region, which is made up of 21 member economies (see Table 1). This region is socioeconomically and culturally diverse, with varying experiences related to mental health support and DMH during the pandemic. Drawing on the perspectives of policy makers, service providers and people with lived or living experience (PWLE) of mental health conditions from across the region, we investigated factors impacting equitable access to and delivery of DMH care and the implications for promoting equitable access by diverse populations to DMH in a post-pandemic Asia-Pacific and beyond.

## Methods

We used a modified Delphi consensus methodology, which is effective for undertaking priority setting activities that involve diverse and geographically widespread stakeholders and when relying on online data collection methods [37–39]. We followed the Delphi process of using previous rounds (both surveys and consultations) to inform subsequent survey and consultation questions in order to increasingly refine the process of priority setting, with the first survey round informed by a rapid review of the literature [1]. We made several modifications to the traditional Delphi technique. In addition to reaching consensus on a topic, Delphi processes can be used for "exploration of a field beyond existing knowledge" [39]. Our intention, therefore, was not to reach full consensus but rather to generate a comprehensive picture of the landscape, needs and priorities related to DMH access among at-risk groups across the region in the context of an unprecedented global health emergency in order to generate recommendations to improve DMH equity. We included three broad expert groups in this process: 1) policy makers, 2) clinicians, care, and service providers; and 3) people with lived or living experience (PWLE) of mental health conditions and their families or caregivers. An additional modification to the traditional Delphi method was to include different experts within these groups throughout the process in an effort to ensure that we captured a diversity of perspectives across a broad geographic area and from participants with varying experiences. Finally, to provide more in-depth insight into the research topic, we included online consultations with expert panels in between survey rounds, drawing on the results of these consultations to inform the ranking process in the final survey, as described below.

### Data collection

We first conducted a rapid scoping review to understand the emerging literature related to mental health equity and access to DMH in the APEC region during the early stages of the

**Table 1. APEC member economies by World Bank income group.**

| LMICs | HICs |
| --- | --- |
| Brunei | Australia |
| People's Republic of China | Brunei Darusallam |
| Indonesia | Canada |
| Malaysia | Chile |
| Mexico | Hong Kong |
| Papua New Guinea | Japan |
| Peru | South Korea |
| The Philippines | New Zealand |
| Russia | Singapore |
| Thailand | Taiwan |
| Vietnam | USA |

pandemic [1]. The results of the scoping review informed the design of a rapid online exploratory survey [40], disseminated amongst members of the APEC Digital Hub for Mental Health, June 16 -July 27 2020 using Qualtrics survey software [41]. This survey provided an initial understanding of the mental health impact and response related to the COVID-19 pandemic, particularly for priority populations and DMH, in the APEC region.

The results of the rapid survey informed discussion questions for online consultations that took place with policy makers, clinicians/ care providers, and PWLE/ representatives of community-based organizations between October 21st and November 16th of 2020. We conducted two sessions for each group to facilitate participation by people in different time zones. The consultations were conducted online using Zoom and were facilitated by first author JKM. Consultation discussions took place in English, ranged from one to one and a half hours, and were recorded and transcribed verbatim. Consultation discussions included the following questions, revised slightly for each participant category: Which populations are most at-risk of negative mental health impacts during the pandemic? Which groups are most at risk of experiencing barriers to accessing DMH care and what are the barriers? What are the delivery-side barriers to delivering equitable DMH support? What resources and actions would help to improve equitable DMH health access?

Based on the results of the rapid scoping review [1] and a preliminary rapid analysis of the consultation transcripts, we developed an additional online survey to allow for broader regional input into the key study questions. The survey was disseminated online between March 22nd and April 16th, 2021. We asked survey participants to respond to questions about the availability of and access to mental health care in the context of COVID-19 in their area. We also asked them to rank, based on the results of the rapid review and consultation, priorities related to the core questions listed above (at-risk populations, DMH access barriers, DMH delivery barriers, resources and actions needed).

The original intention of this study was to include the perspectives of non-English speaking populations by holding country-specific consultations in Chile, Malaysia and Vietnam. As the pandemic situation evolved and demands on Ministry of Health staff became overwhelming, it was impossible for our in-country partners in Chile and Malaysia to conduct this work. We did, however conduct in-country consultations in Vietnam in Vietnamese. These data are not included in this analysis and will be published elsewhere.

## Study recruitment

We used convenience sampling to recruit for both surveys. The initial rapid survey was shared within the Digital Hub's network via email and an online newsletter. An invitation to participate in the second survey was also disseminated among the Digital Hub network via our newsletter, as well as among the investigators' networks by email and widely using social media. Inclusion criteria were broad to allow for diverse participation and required that participants be aged 19 years or older, be able to provide informed consent and reside in one of the 21 APEC member economies. Participants were asked to select their affiliation (see Table 2) and were able to select more than one option as appropriate.

Consultation recruitment included a combined convenience and snowball sampling approach. Members of our study team recommended key stakeholders. We also conducted a search of relevant organizations across the region and sent invitations by email. We asked confirmed participants to share the invitation with their networks and disseminated the invitation on social media (Twitter, LinkedIn). Due to resource constraints, all data collection took place in English.

**Table 2. Survey 2 participant demographics.**

| Country | % | Count |
|---|---|---|
| Australia | 15.1 | 318 |
| Brunei Darusallam | 1.4 | 29 |
| Canada | 9.9 | 207 |
| Chile | 1.4 | 29 |
| People's Republic of China | 4.6 | 96 |
| Hong Kong | 0.6 | 13 |
| Indonesia | 1.3 | 26 |
| Japan | 0.8 | 17 |
| Republic of Korea | 0.6 | 12 |
| Malaysia | 1.0 | 21 |
| Mexico | 0.8 | 17 |
| New Zealand | 0.7 | 15 |
| Papua New Guinea | 0.3 | 7 |
| Peru | 0.2 | 5 |
| The Philippines | 0.2 | 5 |
| The Russian Federation | 0.2 | 5 |
| Singapore | 0.3 | 7 |
| Taiwan | 0.3 | 6 |
| Thailand | 0.3 | 6 |
| United States of America | 59.9 | 1259 |
| Viet Nam | 0.1 | 3 |
| **Total** | **100** | **2451** |
| **Affiliation (choose all that apply)** | **%** | **Choice count** |
| Policy maker | 13.5 | 411 |
| Healthcare care provider: mental health specialist | 23.5 | 717 |
| Healthcare provider: other specialization | 20.8 | 632 |
| Person with lived experience (PWLE) | 10.6 | 323 |
| Family member or caregiver of PWLE | 11.9 | 362 |
| Representative of a community-based organization | 13.6 | 414 |
| Person who self-identifies as a member of an equity-deserving group. | 1.1 | 33 |
| Representative of a private sector organization or business | 4.3 | 132 |
| Other | 0.67 | 24 |
| Total | 100 | 3046 |
| **Gender Identity** | **%** | **Count** |
| Man | 51.2 | 1063 |
| Woman | 36.0 | 747 |
| Transgender/ Non-binary/ Other | 10.3 | 216 |
| Prefer not to say | 2.5 | 52 |
| Total | 100 | 2078 |
| **Type of location** | **%** | **Count** |
| Urban | 66.1 | 1299 |
| Semi-urban / suburban | 25.3 | 497 |
| Rural | 7.9 | 156 |
| Other | 0.6 | 12 |
| Total | 100 | 1964 |

## Analysis

Survey results were analysed using descriptive statistics and, for the second survey, responses have been disaggregated to show responses from low- and middle-income countries (LMICs) and high-income countries (HICs). We conducted Chi-square Tests of Independence using STATA 18 for Windows [42] to examine whether the observed differences in responses between LMIC and HIC respondents were statistically significant. We adopted a consistent approach to managing missing data across both surveys. During our explanatory analysis we found less than 5% missing data. This, coupled with other analyses focusing on the nature and distribution of the missing data led us hold the assumption that the data were Missing Completely at Random (MCAR) [43]. This assumption and the small proportion of missing data led us to use complete case analysis; for any category where responses were missing, we excluded these non-responses entirely from the analysis and did not impute missing values or assign them a default value. This decision was informed by our intention to maintain the integrity and accuracy of the data collected. This approach is supported by previous studies that showed that missing data is inevitable in mail surveys [44], complete case analysis is a common practice in epidemiological studies [45,46], and if the missing data is MCAR then complete-case analysis will result in unbiased estimate [47–49]. This study employs a descriptive approach and as such, no inferential statistical method is employed or inference/generalization is claimed at the population level.

Table 1 displays APEC member economies divided by income category.

Results of the consultation were analysed using thematic analysis [50] using NVivo 12 [51]. Analysis was conducted by first author JKM and co-author SS beginning with immersion in the data by reviewing the transcripts. We then developed a coding frame using a combination of deductive and inductive approaches, with the initial code book developed in alignment with the research questions and interview guide, and additional codes added iteratively during coding. JKM and SS initially coded two transcripts and compared codes for consistency, discussing and resolving any discrepancies based on mutual agreement [52]. After coding all transcripts, we compared coded data and again discussed any discrepancies. We revisited the coded data to identify core themes, which we compared and discussed until agreement was reached.

## Ethics

Primary ethics approval for this study was obtained from the University of British Columbia's (UBC) Behavioural Research Ethics Board [H20-01993]. Many of the study co-investigators indicated that ethics approval from UBC as the lead institution was sufficient given the study involved online data collection from an international sample. However, ethics approval was required and obtained from some partners, including the Malaysia Ministry of Health's Medical Research and Ethics Committee [NMRR-20-2123-56607], the Centre for Addiction and Mental Health at the University of Toronto [109/2020] and the Vietnamese Institute of Population, Health and Development's Institutional Review Board [2020/PHAD/DELPI-01]. Both surveys included an online consent form from which written consent was obtained by requiring participants to select "yes to continue the survey" in order to access the survey. Participants selecting "no" were immediately directed to a page thanking them for their time and were unable to access survey questions. Inclusion criteria included being 19 years of age or older, being able to read and/or communicate in English, providing informed consent, and living in one of the 21 APEC member economies (see Table 1). Participants were informed that their participation was voluntary and that they could withdraw from the survey or skip any questions without penalty.

We obtained written consent from consultation participants by providing them with a consent form by email when they agreed to participate and they were asked to return a signed e-copy before participating in the consultation. At the beginning of each consultation, the facilitator reviewed the details of informed consent and reminded participants that their participation was voluntary and that they could leave the discussion at any time prior to beginning the consultation. Participants were also given the option to engage anonymously by turning off their camera and using an alias. Quotations by consultations participants are denoted below using PM (policy maker), SP (service provider) and PWLE (people with lived experience) and a randomly assigned number to protect the anonymity of participants.

### Inclusivity in global research

Additional information regarding the ethical, cultural, and scientific considerations specific to inclusivity in global research is included in the Supporting Information.

## Results

### Demographics

We received 24 responses to the rapid survey (Survey 1), with respondents representing Canada (n = 5), Australia (n = 4), the Philippines (n = 4), Indonesia (n = 2), South Korea (n = 2), Chile (n = 1), China (n = 1), Hong Kong (n = 1), Japan (n = 1), Malaysia (n = 1), Thailand (n = 1), and Vietnam (n = 1). A majority of participants described their primary affiliation as being with the government or public sector (n = 9) or an academic institution (n = 9), while other affiliations include health care providers (n = 2), the private sector (n = 1), PWLE (n = 1) or other (n = 2). Fifteen respondents identified as male, eight as female and one preferred not to provide their gender.

The online consultations consisted of eight policy makers from Canada (n = 5), New Zealand (n = 2) and Chile (n = 1), seven clinicians/health care providers from Chile (n = 2), Canada (n = 1), the United States (n = 1), South Korea (n = 1), Malaysia (n = 1) and Russia (n = 1), and eight PWLE or representatives of community-based organizations from Canada (n = 7) and the Philippines (n = 1).

We received 2578 responses to Survey 2. We cleaned the data for bots and spam responses. We initially used the built in spam and survey preview filters included in the Qualtrics survey software [41], which identifies likely bot, duplicate and spam responses. We also excluded responses taking less than 150 seconds (three minutes), which we deemed insufficient to properly engage with the survey. 2151 responses were included in the analysis. Survey participant demographics are shown in Table 2. All APEC member economies were represented, although the majority were from the United States (59.9%), Australia (15.1%) and Canada (9.9%). Participants selected their affiliation and were able to choose all options that applied to them. A majority were healthcare providers, either working directly in mental health (23.5%) or in another specialization (20.8%), while 13.5% were policy makers and 10.6% were PWLE. Other groups included family members or caregivers of PWLE (11.9%) and representatives of community-based organizations (13.6%). Half of participants identified as men (51.1%), 36.0% as women, and 10.3% as transgender, non-binary or other. A majority (66.1%) resided in urban areas, while only 7.9% lived in rural locations.

### Availability and access to digital mental health in the context of COVID-19

We asked Survey 1 participants whether there had been an increase in the use of DMH supports in their country since the COVID-19 pandemic began. Of 19 responses to this question, 17 participants selected 'yes', one selected 'no' and one selected 'unsure'. 'Yes' responses were

**Table 3. Populations targeted by DMH supports.**

| | Choice % (choice n) |
|---|---|
| Healthcare workers | 15.1 (16) |
| Youth (12–25 years) | 9.4 (10) |
| People with existing mental health or substance use disorders | 8.5 (9) |
| Women | 5.7 (6) |
| Seniors (over 65 years) | 6.6 (7) |
| Other frontline or essential workers | 6.6 (7) |
| People experiencing domestic violence | 6.6 (7) |
| Migrant workers | 5.7 (6) |
| Children (under 12 years) | 4.7 (5) |
| People with disabilities | 4.7 (5) |
| Indigenous populations | 3.8 (4) |
| Migrants | 3.8 (4) |
| International students | 2.8 (3) |
| People experiencing homelessness | 2.8 (3) |
| Prisoners | 2.8 (3) |
| Other | 2.8 (3) |
| Sexual minorities (e.g., members of the LGBTQ+ community) | 1.9 (2) |
| Ethnic minority or racialized populations | 1.9 (2) |
| Youth in detention facilities | 1.9 (2) |
| Unsure | 1.9 (2) |
| Total | 100% (106) |

provided by participants from Australia, Canada, Chile, China, Hong Kong, Indonesia, Japan, Malaysia, the Philippines, and South Korea. One participant from Vietnam selected 'no', and one from Canada selected 'unsure'.

We also asked Survey 1 participants whether there were DMH supports in their country that target specific at-risk or equity deserving groups. The most common target populations identified in Survey 1 were healthcare workers (15.1%), youth (9.4%), people with existing mental health or substance use conditions (8.5%), as shown in Table 3.

In Survey 2, we asked participants whether they had noticed an increase in DMH services in their region during the pandemic (Table 4). While a majority in both LMIC and HIC groups reported noticing an increase in DMH services (86.2% and 86.0%, respectively), a Chi-square Test of Independence showed that this observed difference is not statistically significant ($\chi^2$ = 0.173, $P$ = .917). This suggests that the perception of an increase in DMH services is similarly widespread among respondents from both LMICs and HICs.

We asked participants who indicated that they have lived or living experience with mental health conditions whether they had accessed DMH programs or services in the last year (Table 5). A majority (79.9%) responded that they had. While a higher proportion of

**Table 4. Have you noticed an increase in DMH services in your region?.**

| | Total % (n) | LMIC % (n) | HIC % (n) | χ2 | P-value |
|---|---|---|---|---|---|
| Yes | 86.1 (1790) | 86.2 (163) | 86.0 (1627) | 0.17 | P = .91 |
| No | 7.0 (145) | 7.4 (14) | 6.8 (131) | | |
| Unsure | 7.0 (145) | 6.3 (12) | 7.0 (133) | | |
| Total | 100 (2080) | 100 (189) | 100 (1891) | | |

**Table 5. Have you accessed DMH services in the last year?.**

|  | Total % (n) | LMIC % (n) | HIC % (n) | χ2 | *P*-value |
|---|---|---|---|---|---|
| Yes | 79.9 (255) | 91.2 (31) | 78.6 (224) | 3.18 | *P* = .20 |
| No | 17.9 (57) | 8.8 (3) | 18.9 (54) |  |  |
| Prefer not to say | 2.2 (7) | 0.0 (0) | 2.5 (7) |  |  |
| Total | 100 (319) | 100 (34) | 100 (285) |  |  |

respondents from LMICs reported accessing DMH services (91.2%) compared to respondents from High-Income Countries (HICs) (78.6%). However, the differences was not found to be statistically significant ($\chi^2$ = 3.18, $P$ = .204).

## Populations experiencing increased mental health risk and barriers to DMH care access

We asked consultation participants to identify populations at high risk of negative mental health outcomes during the pandemic. Participants broadly identified population groups who might have pre-existing or intersecting vulnerabilities or experience social and structural marginalization, such as individuals with pre-existing mental health conditions, LGBTQIA2S + populations, migrants, and seniors as having a high risk of experiencing poor mental health during the pandemic. This is illustrated in the following quotation by a policy maker:

*"I think the main finding from. . .the work that we are doing in communities is that people who previously experienced health and social inequities are the people who are hardest hit. And that's really people who already experience mental health problems, people who are using substances, especially those that live in kind of precarious conditions and are low income or unstable housing or currently unhoused." PM02-P1*

Living circumstances were also identified by participants as a risk factor for negative mental health impacts. Individuals experiencing isolation, such as seniors and people living alone, might have been cut off from social, family and community support. Individuals who were living in challenging or unsafe home environments during stay-at-home orders were similarly identified as high risk, as described by a member of a community-based organization serving vulnerable youth:

*". . .LGBTQ youth, who maybe they're not like, in a super supportive home environment. Um, and like they depend a lot on. . . social services and connection to community [and having] the opportunity to, like, get out of the home and like connect with peers and connect with support workers and that kind of thing. And they don't have that anymore. And kind of being in a situation where they're kind of like stuck at home for long periods of time with parents who maybe aren't affirming or maybe you don't even know, you know, their identity, it's really dangerous. And, it has a huge impact on their mental health." PWLE01-P2*

Consultation participants were asked to identify populations at increased risk of experiencing access barriers to DMH technologies. Several population categories were identified, including people in rural and remote areas, as illustrated in this quotation from a Malaysian service provider:

*"In Malaysia. . .we have seen that, uh, the people that really could not get you know, the digital mental health and telehealth care, are people who stay at the rural areas you know, at the*

*very remote areas. . .they could not get. . .the Internet lines as good as people who stay in the city. So, if we want to provide the virtual mental health over there using the digital technology is going to be very limited access for them."* SP02-P2

People with low socioeconomic status (SES) were also identified as a group that was at higher risk of negative mental health impacts during the pandemic, and which experiences several barriers to DMH access. One barrier among this population is access to devices or a reliable Internet connection, as described in the following quotation:

*"People in poverty who don't even have a computer or a phone. Yeah, I see it all the time in my work, where like, people can't, cause we deliver, of course, all our counseling services virtually, and so many people, they can't even. . . access our services because they don't have a laptop or anything that's required."* PWLE01-P2

Ethnocultural and linguistic minorities were identified as a population that experiences barriers to DMH access. The challenge of providing effective language interpretation is described by a service provider from the United States of America (USA):

*"And the other group [experiencing barriers] is the non-English and non-Spanish speaking population. Unfortunately, it seems harder to have interpreter services while providing any form of virtual care. . .And with mental health you definitely don't want to kind of keep repeating yourself either. And when somebody is sharing sensitive information, um, that added level of interpretation and the barrier to it makes it harder"* SP02-P1

Seniors were also identified as a population that might have difficulty accessing various forms of DMH support, as described in the following quotation:

*"I think older people. . .who are not very tech savvy, who don't know how to use, um, technology, and even if they know how to use basic social media, they don't know how to settle into that modality of talking to someone over the phone about such heavy issues. It's not something that they're very used to"* PWLE02-P1

We also asked Survey 2 participants to rank which populations are most likely to experience barriers in access to DMH care in the context of the COVID-19 pandemic (Table 6). People living in rural or remote areas were selected as the top priority by all participants, with 50.3% of LMIC participants and 37.8% of HIC respondents choosing this as their top ranking. This

**Table 6. Population most at-risk of barriers to DMH care.**

| | Total % (choice count) | LMIC % (choice count) | HIC % (choice count) | χ2 | p-value |
|---|---|---|---|---|---|
| People living in rural or remote areas | 39.0 (627) | 50.3 (79) | 37.8 (548) | 12.88 | *P* = .075 |
| People living with severe mental illness | 17.7 (285) | 17.2 (27) | 17.8 (258) | | |
| Indigenous populations | 9.9 (159) | 4.5 (7) | 10.5 (152) | | |
| People experiencing poverty | 8.2 (131) | 7.0 (11) | 8.3 (120) | | |
| Seniors | 7.0 (113) | 5.1 (8) | 7.2 (105) | | |
| People living with disabilities | 7.8 (126) | 6.4 (10) | 8.0 (116) | | |
| People living in unsafe situations (e.g., people experiencing domestic violence) | 6.7 (108) | 5.7 (9) | 6.8 (99) | | |
| Non-native language speakers | 3.6 (58) | 3.8 (6) | 3.6 (52) | | |

was followed by people living with severe mental health conditions among both LMIC (17.2%) and HIC (17.8%) participants. Indigenous populations were ranked third by HIC participants (10.5%), while among LMIC participants people living in poverty were ranked third (7.0%). These results were not statistically significant ($\chi2 = 12.88$, $P = .075$). In the 'other' fill-in option, populations contributed by survey participants included: migrant workers, children and youth, ethnocultural minorities, and people who are unemployed.

**Barriers to access.**   In Survey 1, respondents were asked to provide free-form responses to identify the most prominent barriers to DMH access in the context of the pandemic. Responses included: a lack of integration and coordination of supports, services and information; mental health-related stigma and low help-seeking; a lack of linguistically and culturally appropriate options; low availability of devices and Internet in rural and remote areas and among people experiencing poverty; lack of technical literacy, especially among seniors; negative perceptions about the quality and effectiveness of DMH options; concerns about privacy and data security; unprepared health systems and lack of political will.

We also asked consultation participants to identify the access barriers to DMH services faced by at-risk populations. Participants identified access issues such as cultural and linguistic appropriateness as often mental health programs follow a Western model and do not reflect the cultural beliefs or languages of diverse populations. This was identified as a challenge for providing appropriate care to diverse cultural groups, including Indigenous populations, as described by a participant from the Philippines:

> "Yeah, as I see it here in the Philippines. . .the Indigenous people who live in the mountains, remote areas, not only because they don't have a phone, but because of language, language, expression. . ." PWLE02-P2

A lack of accessibility in DMH interventions was also identified by participants as a barrier, including for individuals with disabilities who might rely on non-verbal means of communication, which can be hindered through virtual care or might require the support of a family member, as illustrated by the following quotation:

> "We have individuals that are unable to communicate via, you know, Zoom conversations or whatever, there's a lot of nonverbal things. My daughter uses twitches of her arms, a lot on top of, on top of words, but you need to know that, that she's speaking nonverbally as well, so appointments with her over the computer don't really work. You have individuals with hearing impairments that may not have the right devices to do an online meeting and those that just can't log in without their support worker or family member right beside them to use the devices. So, there is an accessibility issue there for people or those that just can't sit for an appointment online because it has no real value to them." PWLE01-P4

Consultation participants also indicated that access to DMH often varies by the modality in which the care is delivered depending on the specific needs of a population. For example, some types of interventions may be challenging to deliver online, as described by a service provider:

> "So let's say someone with schizophrenia. . .they go during the day to have activities with other with other patients and so that they can be occupied and have social interaction. I'm not sure that that can actually be done online, and maybe they'll have a short visit or something. But there are some things that you can't really, that you can't really do online, I think." SP01-P1

Lack of privacy can be a substantial barrier to DMH access. As previously illustrated, youth with diverse sexual and gender identities, for example, may be at risk if accessing care within proximity of unsupportive families. Privacy is also a potential barrier for people living in small spaces, particularly when balancing mental health care with caregiving obligations. This participant describes a scenario that is common among their clients:

*". . .for dealing with mental health issues, we also have the problem of privacy. Uh, for example, I've got an individual who has undergone some huge mental health crises, a single parent, two children in the house. How do you make your mental health appointment and speak about all of the things and some of them involve the children, when you have to also watch the children and be present and not upset the children because the children have their own mental health things going on? And likewise, the children can't have the virtual mental health because there's three people in a two bedroom house or a two bedroom apartment, one laptop. So who's doing school? Who is able to talk to the counselor? How do you navigate being able to be open and transparent when somebody is always 10 feet away?"* PWLE01-P4

Concerns regarding quality and evidence base of DMH interventions are also a core theme related to access barriers. Due to the need for DMH during the pandemic, many services pivoted to DMH with little to no evaluation of the quality of care provided, as illustrated by the following quote from a service provider:

*"As in Malaysia, we have so many NGOs. . .or societies or associations, that provide for mental health services through uh, telephone to, you know what, uh, any other social media intervention. And most of the time we could not control them, the content that they are giving to the society, the kind of services, we cannot control them. So, we are worried actually, in terms of the effectiveness of the services they provide, because we know that what is happening now is something that uh, we did not have time to prepare earlier on. So whoever has the facility, whoever has the time to provide services they just do. We do not have the proper training, as [we do] for our, you know, for our government and psychologist and counselor as well. . .honestly speaking, I still do not have 100 percent confidence in terms of the effectiveness of the services."* SP02-P2

The final access barrier that was highlighted is building trust and therapeutic alliance. Participants described the challenges of creating a trusting relationship between a patient and a therapist virtually, compared to in-person. A policy maker from Canada describes the issue of trust in the context of developing new therapeutic relationships online:

*". . .mental health care often is correlated to kind of trusting relationships between the recipients of care and the practitioners, and cultivating trust through virtual means is much more difficult for individuals who have preexisting conditions and have had traumatic experiences accessing health care and don't have established relationships with providers, it would be very difficult for them to reach out and establish those relationships now through virtual means when they have no other way of connecting with that person."* PM02-P1

Survey 2 participants also ranked the most important barriers to accessing DMH care (Table 7). Over half (52.2%) indicated that limited or no Internet connectivity was the highest barrier, with 59.9% of LMIC and 51.3% of HIC participants selecting that as their first choice. Among LMIC participants, cost of digital mental health programs or services (7.9%) and limited access to devices (5.9%) were ranked second and third. Among HIC participants high cost

**Table 7. Barriers to accessing DMH care.**

| | Total % (choice count) | LMIC % (choice count) | HIC % (choice count) | χ2 | P-value |
|---|---|---|---|---|---|
| Limited or no Internet connectivity | 52.2 (796) | 59.9 (91) | 51.3 (705) | 16.35 | *P* = .23 |
| High cost of accessing Internet | 11.0 (168) | 5.3 (8) | 11.6 (160) | | |
| Cost of e-mental health care programs or services | 6.2 (95) | 7.9 (12) | 6.0 (83) | | |
| Lack of culturally appropriate care options | 4.7 (71) | 2.6 (4) | 4.9 (67) | | |
| Limited access to devices | 4.6 (70) | 5.9 (9) | 4.4 (61) | | |
| There are so many e-mental health options people don't know where to start | 3.7 (57) | 2.0 (3) | 3.9 (54) | | |
| Low technological literacy | 3.1 (48) | 2.6 (4) | 3.2 (44) | | |
| Closure of public spaces to access Internet and computers | 2.2 (33) | 0.7 (1) | 2.3 (32) | | |
| Difficulty building trust between patient and provider with online options | 2.5 (38) | 1.3 (2) | 2.6 (36) | | |
| Low mental health awareness among people in need of care | 2.1 (32) | 1.3 (2) | 2.2 (30) | | |
| Lack of private space to access care | 2.0 (31) | 2.6 (4) | 2.0 (27) | | |
| Limited availability of programs or services in multiple languages | 2.0 (30) | 2.6 (4) | 1.9 (26) | | |
| Stigma preventing people from seeking help | 1.9 (29) | 2.6 (4) | 1.8 (25) | | |
| Uncertainty about how effective e-mental health care is | 1.8 (28) | 2.6 (4) | 1.7 (24) | | |

of accessing the Internet was ranked second (11.6%) while cost of digital mental health program was third (6.0%). However, as shown in Table 7, there is no statistically significant difference between the responses from LMIC and HIC participants (χ2 = 16.35, *P* = .23) Additional responses in the 'other' category include privacy concerns and concerns about the appropriateness of digital mental health for people with serious mental health conditions.

**Barriers to digital mental health delivery.** Consultation participants were asked to identify barriers to equitable DMH from the delivery side, including from the perspective of healthcare providers and health systems. We identified several themes based on the responses. First, and consistent with the themes described above, are challenges related to providing culturally and linguistically appropriate care. One such challenge is the capacity of care providers and organizations to provide appropriate care to meet the needs of diverse patients or clients. These challenges are described by a representative of a Canadian community-based organization:

> "...as we explore these different ways to be more accessible and better serve different vulnerable populations, we come up against capacity issues all the time. So there's issues of like if we try to expand to include a different language, how do we hire a workforce of folks who speak that language or have that cultural competency or come from that background when we also have the volume of only...it's a marginalized population or a minority population." PWLE01-P1

Trust and the ability to reach ethno-culturally diverse communities was also raised as a challenge, as described in the following quotation:

> "...we heard from various stakeholders was that a young person will not trust a service that they haven't heard of from one of their peers or one of their community leaders, so that outreach has to come from their community or from their local mosque or community centers...It does take a lot more time and planning, and we're fortunate that we were able to do most of those, um, relationship building cases before the pandemic happened. But I can't

*imagine what that would have looked like if we were to do it, um, in the current times."*
*PWLE02-P1*

Digital literacy and access to the necessary devices among providers was also identified as a challenge faced in the context of the pandemic, as described below:

*"...when we had to transition away from all being at the office and to like working at home, there was sort of this assumption that like, well, everyone has what they need, like everyone has all the time that they need to now, like do all of their work from home. And there's a huge amount of privilege in kind of just assuming that, like all your staff are going to have a stable Internet and phones and computers and all of these things." PWLE01-P2*

Concerns among providers about various aspects of DMH, ranging from privacy and confidentiality, self-efficacy, the effectiveness of DMH and the therapeutic alliance were also identified as barriers, as described by a Canadian service provider:

*"...the most significant challenge for mental health is resistance from the mental health providers, and not resistance because they don't want to, resistance because they're afraid of lack of confidentiality, that we won't be able to provide the strong bond. And interestingly, unfortunately, the data have been there for years that it is doable, safe and effective. Uh, so I've been working a lot since March to provide workshops. Essentially, I'm giving a workshop every week almost to mental health care professionals to debunk the myths that everything will go wrong." SP02-P3*

Workload and workflow in the context of the pandemic was also identified as a challenge of providing DMH care from the perspective of providers. Many described the challenges associated with working from home and providing care online, as described by this service provider:

*"...But now I have the challenge of [working] at home with my kids around. So I try to lock myself in a room. It doesn't always work, but I always have, I'm concerned that the patient will be able to hear what's going on at home. A few times one of my children has appeared on the screen, which I don't mind in a regular meeting, but I do mind it a lot with a client. And then also sometimes connection is not that good and the patient is talking about things that are very difficult for the patient. And then you have to ask them to repeat or you have to connect again." SP01-P1*

From a health systems side, a lack of available guidelines and best practice frameworks for providing DMH care posed a challenge when service provision switched rapidly away from in-person appointments to online or tele-health approaches, as described below:

*"...we have so many different platforms for like delivering services, right? We have Zoom, we have Doxy, we have... I'm sure that there's way more that I like, in this moment I can't think of, but there's many different ones, right? And so when my work had to transition to providing counseling services online, there were all these questions of like, well, which one do we use in terms of, like, confidentiality, right? ...And like there were a lot of different like conversations and confusion around confidentiality with, with these actual platforms and...like from our perspective as service providers, it was really confusing to know, like what is compliant, what isn't, you know?" PWLE01-P2*

Communication, awareness and referral pathways were also identified as challenges from the delivery side. One Canadian policy maker stated:

*"So with the first [challenge] around quality, we heard just so often. . .that in terms of prescribing different, um, applications, and so I'm not talking about the telehealth side, I'm talking about specifically about programs online, just uh, not knowing what is based on evidence, what is going to work for a particular audience, age group, etc. . ."* PM02-P3

Finally, fragmentation across systems was also identified as a challenge, which can negatively impact patient pathways and consistency of care. A policy maker from New Zealand stated:

*"So if I use New Zealand as an example, if I'm in a hospital system, and my EMR system, and if I'm in a primary care, same patient, but I go and see my GP, that the patient management system, and the other two shall never ever talk with just the way the procurements have historically happened. They are different systems. There's no integration. So the patient journey, care pathways, it's um, the patient is having to repeat [their] story all the time. And sometimes they would ask you a very simple question: don't you guys talk to each other? In this day and age in 2020, we are still not being able to share the data at the right point of time."* PM01-P4

We also asked Survey 2 participants to rank the most significant challenges in the delivery of DMH services (Table 8). Concerns about building trust and a therapeutic alliance with patients was rank highest (51.2%), with 62.9% of LMIC and 49.9% of HIC selecting that as the most important challenge. Provider difficulty in picking up non-verbal cues from patients such as body language was ranked second among both LMIC (8.6%) and HIC (15.3%) participants. Among LMIC participants, both lack of access to technology and infrastructure by providers and logistical challenges such as referral pathways were ranked third (5.3% each). HIC participants indicated that concerns about privacy, security and confidentiality (7.8%) was the third most important challenge. These results were not statistically significant ($\chi2$ = 14.42, *P* = .15).

**Table 8. Barriers to DMH delivery.**

| | Total % (Choice count) | LMIC % (Choice count) | HIC % (Choice count) | χ2 | p-value |
|---|---|---|---|---|---|
| Providers worry about building trust with their patients (e.g., therapeutic alliance) | 51.2 (760) | 62.9 (95) | 49.9 (665) | 14.42 | *P* = .15 |
| Providers find it hard to pick up non-verbal cues from their patients | 14.6 (217) | 8.6 (13) | 15.3 (204) | | |
| Providers have concerns about privacy, security and confidentiality | 7.5 (111) | 4.6 (7) | 7.8 (104) | | |
| Lack access to necessary technology or infrastructure by providers | 5.9 (87) | 5.3 (8) | 5.9 (79) | | |
| Providers find it more difficult to deliver care using e-mental health (e.g., Zoom fatigue, feeling isolated) | 4.0 (60) | 2.6 (4) | 4.2 (56) | | |
| There are logistical challenges with e-mental health care (e.g., referral pathways, prescriptions, coordination of electronic medical records) | 4.0 (60) | 5.3 (8) | 3.9 (52) | | |
| Healthcare providers might not be aware of e-mental health programs or services | 3.2 (47) | 1.3 (2) | 3.4 (45) | | |
| Providers have limited training in providing e-mental health care | 3.2 (48) | 4.0 (6) | 3.2 (42) | | |
| Providers might not be aware about which e-mental health programs and services are effective (e.g., which are evidence-based) | 3.0 (45) | 2.6 (4) | 3.1 (41) | | |
| There are no available guidelines for how to deliver e-mental health care | 1.8 (27) | 1.3 (2) | 1.9 (25) | | |
| Providers feel the move to e-mental health was too fast and it is hard to keep up | 1.4 (21) | 1.3 (2) | 1.4 (19) | | |

**Resources and actions needed.**   Numerous themes were identified from the participants' responses regarding what resources and actions are needed to increase access to DMH for at-risk populations. The first is affordability, with participants discussing the costs associated with accessing mental health care, including in health systems such as the United States where access to care can be unaffordable to many. Even in publicly funded health systems, cost barriers often exist. A policy maker describes this challenge in the Canadian context:

*". . .one thing that I'd like to point out is that here in [province], not all mental health care is publicly funded and publicly accessible. A lot of mental health care is privatized. So making a greater range of services available to people free at point of use, because right now we see programs that are free for mild to moderate symptoms. . .and then it's, people can access our public health care system to see a psychiatrist or be admitted into intensive treatment. But that middle ground of, of counseling and different forms of evidence based therapies is still paid out of pocket or through insurance. And for a lot of, um, at risk groups that may be living in poverty, it's just completely inaccessible. So thinking through how we can upscale access through public funding"* PM02-P1

Participants also described the need for partnerships with the private sector to support improved access to devices and Internet infrastructure. One participant describes the efforts of First Nations communities in Northern Canada to ensure access to devices, calling on private sector telecommunications companies to do the same:

*". . .one of our colleagues who works in the north, where she lives, their local [First Nations] band supplied all the kids with tablets not that long ago into COVID. So they had access. And so, you know, let's call it as it lays. The haves have the ability to provide for the have nots, meaning the big companies like Telus and Bell, and they can throw WIFI anywhere they want during this. And I haven't seen it. But it's such a simple thing, it's such a simple thing."* PWLE01-P3

Participants also highlighted the importance of collaboration, coordination and learning across sectors, whether between different organizations, the private and public sectors within the same country, or between different countries:

*". . .there are so many resources out there, but not enough ways to connect them to the right people, so we, at our organization, we have an information and referrals database, many other organizations have their own databases through which you can find resources based on the category that you're looking in. Um, but they're in pockets, in different regions, in different provinces and catering to different communities. So I feel like an integrated overall hub would go a long way if that's something that can be achieved. Um, I also feel like cross industry collaboration, so, for example, the newcomer settlement sector collaborating with. . .the doctors who often need to provide referrals to patients collaborating with, um, the government sector. So various sectors need to be more cohesive in terms of what they're providing to the at risk populations."* PWLE02-P1

Another theme related to resources and actions includes raising awareness and communication regarding the evidence supporting DMH approaches, ensuring that providers and patients are aware that many DMH approaches have a high standard and are often as effective as in-person mental health supports:

*"We ran a large scale project in [Canadian province] around e-mental health, and it was a mix of in-person and e-solutions. But, you know, the number one thing we learned when we were asking patients and practitioners around, why didn't you use e-mental health?. . .It was really around, not only did they not know the services existed, and/or the quality of the evidence, the number one thing was that they thought it didn't work as good as in-person for whatever they came in for. And so it's also some myth busting and depending on what the cause was of why someone is reaching out to services in the first place."* PM02-P3

The aspect of choice and agency for patients accessing care should also be considered. For example, some individuals prefer face-to-face care as opposed to virtual care. Other patients might have a preference when it comes to choosing specific institutions to receive their care from to support their privacy. One PWLE participant describes these considerations:

*"Yeah, just kind of on the opposite, um, on the comment, you know, everybody prefers in-person care. So, I think on the whole that that's probably true. But for some people, it's not true. Like, I'm service provider [and] service user. And so since this is all started, I've been able to access my care through Zoom or Skype, and that actually protects my privacy because my service provider is at the same institution where I provide care and I always risk running into people who I might be providing care for. So since this started, I've actually wished that I would have had this a long time ago because it's a big peace of mind for me."* PWLE02-P3

A policy maker further underscores the role that DMH must play within the broader mental health system to ensure that care modalities are appropriate for all who need them:

*"And so it's really about the fact that virtual health is part of a system and that it should be up to the individuals based on recovery oriented principles and choice where they would like to obtain their services. And often for most people, it's probably blended."* PM02-P3

Capacity building in the use of digital technologies is also needed, both for patients and providers. For example, training opportunities for seniors can help them to access technology. Healthcare providers may also need to be trained to using digital services as well as to appropriately support specific vulnerable populations as described by a service provider from Chile:

*"Yeah, in my case, in the transgender population, we need different resources. Uh. . .technological resources, but also the capacity of the, the providers, like very specific training,. . .because our patient needs to feel safe with the therapist. And. . .in Chile, at least, um, few places who have care services, um, have a professional who [have] high level of training for working in therapy or in psychiatry with the transgender people."* SP01-P2

Finally, participants agreed that having an established framework to work on DMH to serve the public healthcare systems across countries would be very beneficial. One policymaker used Australia as an example of best practice in this arena:

*". . .So, so first and foremost, I would like to see that we work towards having some e-mental health framework for our countries, some framework which uh, allows us to do the procurement in the right place, at least on with our federally funded initiatives. Private people can do whatever they feel like. But as a nation, we must have a e-mental health framework. Um, recently, Australia has gone down the path of developing e-mental health and addiction standards. And the Care and Quality Commission has developed those. . .It has been a multi-year,*

*massive piece of work and a beautiful piece of work. So e-mental standards have been developed for Australia. . ."* PM01-P4

Regarding resources and actions needed to improve equitable access to DMH programs and supports (Table 9), both LMIC (62.0%) and HIC (50.8%) participants ranked providing culturally appropriate and safe programs and services as the most important. HIC respondents (14.6%) ranked providing DMH services in multiple languages as the next important, while LMIC participants (6.0%) indicated that making Internet or device access free or low cost was the second most important. Providing targeted communications to at risk groups about DMH services was ranked as third most important by HIC participants (6.8%) while providing multi-lingual options was third among LMIC respondents (5.3%). We identified a statistically significant difference between HIC and LMIC responses, as shown in Table 9 ($\chi$2 = 38.04, P < .001).

**Opportunities.** Although this study predominantly focused on challenges and next steps, several opportunities were also identified by consultation participants related to the shift to DMH during the pandemic. First, the proliferation of DMH approaches during the pandemic was seen as a substantial learning opportunity, including across the Asia-Pacific region and across health sectors, as described by one policy maker:

*". . .the connectivity and the penetration of connectivity in developing countries is, is phenomenal now. And there are many health programs. I worked a lot on tuberculosis, HIV, malaria, etc., are using digital platforms for following up on treatment, compliance, etc. on that. . .So one thing that I would say it would be useful to look at, how can digital mental health ride on the successes of other digital uh, platforms in health, and how can it succeed better?" PM02-P2*

Despite many challenges related to equitable access, the rise of DMH during the pandemic was also seen as an opportunity to improve access to care by underserved or equity-deserving populations. Participants indicated that the shift to DMH helped to facilitate access by people experiencing barriers related to location, including people in rural and remote areas and

**Table 9. Resources and actions need to improve equitable access to DMH.**

| | Total % (Choice Count) | LMIC % (Choice count) | HIC % (Choice count) | χ2 | p-value |
|---|---|---|---|---|---|
| Providing culturally appropriate and culturally safe programs and services | 51.9 (795) | 62.0 (93) | 50.8 (702) | 38.04 | *P* < .001 |
| Providing programs and services in multiple languages | 13.7 (210) | 5.3 (8) | 14.6 (202) | | |
| Providing access to free programs and services | 7.6 (117) | 4.0 (6) | 8.0 (111) | | |
| Improving targeted communication about available programs and services to increase awareness among at-risk populations | 6.3 (96) | 1.3 (2) | 6.8 (94) | | |
| Making devices and Internet access available for free or at low cost | 4.1 (63) | 6.0 (9) | 3.9 (54) | | |
| Improving Internet access and availability in rural and remote areas | 3.7 (56) | 3.3 (5) | 3.7 (51) | | |
| Developing clear guidelines for policy makers on e-mental health care delivery | 2.8 (43) | 4.7 (7) | 2.6 (36) | | |
| Developing partnerships with community-based organizations | 2.4 (36) | 4.7 (7) | 2.1 (29) | | |
| Co-designing programs and services with people who will use them | 2.1 (32) | 2.0 (3) | 2.1 (29) | | |
| Providing increased training, support and supervision for providers in DMH | 1.7 (26) | 1.3 (2) | 1.7 (24) | | |
| Developing clear guidelines for healthcare providers on DMH care delivery | 1.6 (25) | 4.0 (6) | 1.4 (19) | | |
| Improving referral pathways and navigation of care options for patients | 1.1 (17) | 0.0 (0) | 1.2 (17) | | |
| Providing training or support to improve technological literacy for service users | 0.8 (12) | 0.7 (1) | 0.8 (11) | | |
| Developing partnerships with private-sector organizations (e.g., telecom companies) | 0.2 (3) | 0.7 (1) | 0.1 (2) | | |

people for whom transportation is a barrier due to cost, childcare needs or other circumstances. DMH can also help to advance access to mental health care in LMICs, where existing mental health supports are often scarce. DMH has also enabled access by populations who might experience barriers to mainstream care, as described by a service provider from Chile:

*"Yes, in my case with transgender people and in our service, this pandemic has been an opportunity too, because in Chile, the resources of the mental health care with this kind of population is in, in few places in Chile with very difficult to find safe places for mental health care. So the psychological attention by, by online has been an opportunity for that, at the same time of the risk. It is very interesting moment in that case, probably when, when we back to the face to face, we keeping this online service for that kind of young people who found, in the, where they living, um, some professional care or safe place for, for attending the psychological problems."*
*SP01-P2*

The advancement of DMH within the context of the pandemic was also seen as an opportunity to support mental health system strengthening, including via the increased adoption of DMH technologies beyond the pandemic itself. One policy maker states:

*"Now, I would look at this as an opportunity. Even pre-COVID, access to mental health we know around the world is a major problem. There are problems of health systems, stigma, accessibility, acceptability and culturally specific norms in different countries. . . .I would look at this as an opportunity that beyond the COVID-19 pandemic, how can digital mental health and telehealth care complement the already [existing] health systems that are rolling out mental health services in different settings and countries?"* *PM02-P2*

Finally, the need for DMH to promote mental well-being through connection and to avoid the risk of DMH leading to further isolation is imperative, as described by a policy maker:

*". . .and one of the things that I think that the pandemic's brought to light is the importance of social inclusion and connection for mental health and overall sense of well-being. There's been many risk factors with regards to the pandemic that have increased the incidence of mental health and substance problems. But I think social isolation has been a really big one. And I think in terms of virtual mental health care is we really need to think about the ways in which it can provide a source of connection and help break that isolation, but at the same time reinforce it, and be really, really attentive to those distinctions, especially with regards to different groups who are accessing these services with always keeping in mind that mental health care should be fostering connection and support, not isolation."* *PM02-P1*

## Discussion

This mixed methods study has captured the perspectives and priorities of people living across the APEC region related to DMH access among vulnerable and at-risk populations during the COVID-19 pandemic. Study participants include people with lived and living experience of mental health conditions, their families and carers, clinicians, community and social service providers, policy makers and others living in both HICs and LMICs across the region. The results, reflecting the experiences of stakeholders during the first year of the pandemic, provide insight into challenges related to mental health risk and DMH access during what may be considered the most difficult time of the pandemic. They also offer lessons learned that can inform recommendations to promote equitable access to DMH across the Asia-Pacific and beyond in

**Table 10. Summary of results and recommendations.**

| Challenge | Recommendation(s) |
|---|---|
| Gap in evidence based DMH interventions for serious mental illness (SMI) | Advance research and policy to support access for people with SMI, especially in LMICs |
| Diverse populations underrepresented in DMH design, research and policy development | Integrate equity considerations into DMH research, monitoring and evaluation; Meaningfully engage diverse people with lived experience in all facets of DMH research, design, development and policy processes. |
| Privacy and safety concerns as an access barrier for people in challenging living situations | Include non-verbal options for engagement and built-in safety mechanisms |
| Low digital literacy among some populations | Include training, support and accessibility features to promote ease of use and increased digital literacy |
| Lack of digital infrastructure (e.g., rural and remote areas) and device access (e.g., low SES populations) | Increase public and private sector investment in digital infrastructure and access, including via public-private partnerships |
| Need for improved DMH governance and policy | Integrate DMH training into curricula; Develop, implement, and disseminate clear and transparent data privacy and security policies; Extend payment structures to include coverage for DMH services |
| Lack of trust and confidence in DMH | Promote transparent, targeted, and accessible communication about DMH effectiveness, risks and availability |
| Lack of standards for DMH dissemination, health system integration and delivery | Create and implement policies, standards, and frameworks with explicit equity considerations |
| DMH equity interacts with structural and systemic challenges (e.g., mental health system capacity, social and digital determinants of health) | Take opportunity to invest in comprehensive, equity-oriented policies that promote equitable DMH within strong mental health systems; Collaborate across sectors to promote progress on social, structural, and digital determinants of health |

the long term, as health systems increasingly invest in DMH options and hybrid care models become the norm [53]. Results and recommendations are summarized in Table 10.

The results of this study reflect the proliferation of DMH options early in the pandemic, with a large majority of survey participants in both HICs and LMICs noticing an increase in DMH supports. Though results were not statistically significant, a higher proportion of LMIC respondents indicated they had accessed DMH supports compared with their counterparts in HICs. Though a large majority of respondents were from HICs, responses from LMICs might suggest that DMH may have been particularly welcome in contexts with limited mental health system capacity and low availability of mental health care. In LMICs during this time however, a majority of DMH options targeted general distress or mild to moderate common mental health conditions, meaning that people living with serious mental illness often lacked access to the necessary supports [54]. This reflects a pervasive gap in evidence-based care for people living with serious mental health conditions in LMICs [55]. It is therefore imperative to ensure that research and policy to advance DMH in LMICs includes care for a spectrum of mental health conditions.

It is evident that during the pandemic pre-existing vulnerabilities and barriers to a care access were exacerbated. It is also clear that a one size fits all model of DMH care perpetuates the digital divide in access, use, appropriateness and quality of DMH care for at-risk and vulnerable populations. For example, providers described the added challenge of offering culturally and linguistically appropriate care to ethnoculturally diverse patients using DMH approaches, including difficulties using interpreters and lack of culturally or linguistically appropriate resources. Cultural and linguistic appropriateness was also identified as a major

access barrier by service users in both the survey and consultations, with many DMH options drawing only on Western concepts and approaches and often not available in Indigenous or ethnocultural minority languages [36].

The disproportionate negative health and social impact of the pandemic on racialized and ethnocultural minority populations has been described elsewhere [1,56–58] as has the frequent exclusion of these populations from research to design and test DMH interventions [59]. Steps must be taken, therefore, to ensure DMH interventions promote rather than impede access by those most in need. Friis-Healy et. al [56] advance recommendations to make DMH interventions more accessible to racialized and ethnocultural minority populations. Their recommendations include engaging in innovative randomized trial methodologies that assess the effectiveness of interventions when they are adapted for diverse patient characteristics, ensuring diverse populations are included in implementation science and user-centered design research, and engaging in outreach to build trust and confidence among racialized and ethnocultural minority communities. These principles could also be applied to enhance the likelihood that interventions meet the needs of other vulnerable populations including sexual and gender minorities, people with disabilities and people with serious mental health conditions.

Several additional DMH access barriers were identified in this study, including lack of privacy, and living in small spaces. For example, this was identified as a substantial barrier for youth with diverse sexual and gender identities who may be living in unsupportive or non-affirming situations, people experiencing domestic violence, or people with low SES [1]. This is consistent with studies showing that upon switching to several forms of DMH services, individuals who benefited the most included those having a private or safe space [36,60–63]. This should therefore be considered in the design of DMH interventions to allow service users to engage safely. Steps to do so could include ensuring programs such as apps are password protected, providing text-based options such as direct messaging that do not require that user to speak out loud, and emergency exit buttons that allow users to quickly close programs on their devices [64].

Digital literacy was also identified as a barrier to accessing DMH especially among individuals with pre-existing health inequalities [65]. For example, elderly populations may experience technological literacy barriers, meaning they may not be able to utilize DMH services on their own and may have to rely on family members or other caregivers to help navigate new technologies [35]. Seniors may encounter numerous barriers when it comes to engaging with DMH services such as a lack of guidance or instructions, lack of confidence or knowledge, a lack of social interaction and distrust in the use of technologies [35,66]. Based on experiences with DMH during the pandemic in Los Angeles, Ojha & Syed [35] recommend strategies to address such barriers for seniors including creating step-by-step guides, having options to change font size and using clear color differentiation and text supported with picture representations. They further suggest implementing a 24/7 service line to help seniors with any technological problems they might encounter.

Access to internet and digital devices was rated as a top barrier to accessing DMH services by survey participants in both LMICs and HICs, particularly among rural and remote populations. Lack of access to a device (such as smartphone, laptop, or tablet) and internet connection (cell phone data or wireless), has been identified as a major barrier to accessing DMH services across the Asia-Pacific region in both HIC and LMIC contexts [36,67]. The gaps in access experienced by rural and equity-deserving populations including Indigenous communities in Canada [30], China [31] and Indonesia [32], for example, were described above.

Several studies presented immediate strategies to address this gap during the pandemic, such as providing families with temporary devices and internet access or assisting them in accessing DMH services through leveraging auxiliary staff or community health workers [67–

69]. To ensure sustainable improved access to digital technologies there is an urgent need for more long-term government and private sector investment in rural and remote technology infrastructure and to support access to devices among underserved populations. While governments play a central role in expanding technological access, the onus does not just lie in the health sector. Private companies, especially in the technology sector, can play a substantial role when they partner with health systems to improve access to digital technologies, improve infrastructure, and address the social determinants of health and technology access that may act as barriers for equity-deserving groups [70]. A review of technology sector-led initiatives to address digital health equity [70] identified several programs to support access to education, transportation, housing and other supports. There is great potential for these types of programs, when conducted ethically and with transparency [71], to have a real impact for access to DMH technologies by providing targeted supports to equity-deserving populations.

The results of this study also show that a common experience for providers, PWLE and policy makers across the Asia-Pacific during the pandemic was uncertainty about many aspects of DMH, including its effectiveness, availability, quality, implementation and safety. This emphasizes the need for governance and policy structures that support equitable delivery of DMH care by mental health systems in a way that provides clarity and confidence to all stakeholders. Several aspects of DMH governance and policy should be considered. As described above, many providers were uncertain about how to deliver DMH care and about the effectiveness of these interventions. Training is therefore essential to ensure providers have the appropriate competencies to deliver care using digital or telehealth modalities. Training and supervision in DMH should be integrated into standard mental health, primary care, and other relevant training programs [35,72]. Privacy is another major concern related to DMH, and policies that promote data privacy and security are essential. This includes regulations such as PIPEDA in Canada and HIPAA in the US [72,73]. Payment structures must also be extended to ensure that DMH interventions are covered similarly to in-person care by both public health systems and by private insurers. Ensuring payment structures include DMH care options can in turn promote access to evidence-based care and helps to avoid reliance on ineffective or potentially harmful options that may be accessed for free online [28,35].

Trust and confidence are also key considerations for DMH equity at the individual level, with building trust and a therapeutic alliance between service users and providers identified as a substantial barrier by survey participants. It is common for individuals, especially from underserved communities, to lack trust and confidence in DMH options [74]. To address this challenge, transparent self-monitoring of DMH needs to be promoted in the industry, in addition to conducting research and communicating results about the effectiveness or DMH interventions with both providers and patients [56]. For example, despite concerns about establishing therapeutic alliance through DMH services [65], a 2017 review [75] found that therapeutic alliance ratings of several digital health services were similar to those of face-to-face therapies. Consumers must also be empowered to assess potential risks, such as data security, which could be communicated through simple, concise statements or warnings using plain, accessible language [74]. Achieving this will require the development and enforcement of transparent privacy and data security policies, requiring an open dialogue between all stakeholders, including service users, providers and DMH companies [56]. Gordon et al. [76] suggest the integration of clinically supported DMH products into electronic health records and patient portals as a means to disseminate DMH products such as apps into clinical care in an efficacious and safe way. This approach would empower and educate both providers and consumers to seek effective DMH care choices.

Guidelines and frameworks are thus needed to support health systems to implement DMH in a way that is consistent with standards of best practice [77]. The Mental Health Commission

of Canada, for example, developed a *Toolkit for e-Mental Health Implementation* [78] as a resource for providers, managers and policy makers. As described above by a policy maker participant, the Australian Commission on Safety and Quality in Healthcare developed *National Safety and Quality Digital Mental Health Standards* [79]. These initiatives present an opportunity for sharing knowledge and best practice related to DMH across the Asia-Pacific region, particularly with LMICs where, in many cases, DMH is still emerging. Policies, guidelines, and frameworks must also include considerations for the equitable delivery of DMH to ensure that the most vulnerable populations are not excluded from accessing care [1,34,74].

Though many specific barriers and risk factors can be identified that influence access to DMH care, access is ultimately influenced by numerous complex and interconnected factors. To understand and address this complexity, Crawford & Serhal [34] introduced the Digital Health Equity Framework (DHEF), which conceptualizes digital health access in relation to the digital determinants of health, suggesting that equitable access to digital health technologies is interconnected with individual, social, cultural, and economic factors and to the social determinants of health [34]. They also argue that digital health equity requires not only equitable *access*, but also equitable *outcomes*, meaning that everyone must be able to benefit from accessing digital health by ultimately experiencing improved health outcomes. To improve digital health equity, ensuring that research and evaluation of digital health technologies includes explicit equity-oriented data to measure access and outcomes by diverse populations is therefore essential [34,56]. It is also crucial for people with lived and living experience of mental health conditions, representing diverse communities, to be integral partners in all facets of DMH research, design, development, evaluation, and policy, and to be represented in leadership and decision-making roles within these initiatives [1,34].

The results of this study demonstrate that, in many cases, those with poor access to mental health care also experience poor access to DMH. The acceleration in the use of DMH during the pandemic took place in the context of pervasive mental health system challenges, including long wait times and high out-of-pocket costs in HICs [80,81] and extremely limited resources and care availability in LMICs [7]. Building on its use during the pandemic, DMH has the potential to improve access to care among many populations in both HICs and LMICs. The expansion of DMH, however, must take place in a way that considers equity in access by diverse populations. It must also take place in the context of broader and long-term mental health system strengthening that prioritizes universal access to all types of care in a way that is patient-centered and responsive to the needs of people with mental health conditions.

## Limitations

We used a non-probability convenience sampling approach for survey recruitment, which limits generalizability of the results. We were, however, able to recruit a substantial sample that included responses from across the APEC region and by all stakeholder groups. The mixed methods approach, which included consultations with the three broad stakeholder groups of interest, further strengthen the results by provided in-depth perspectives related to the issue of digital mental health equity across the APEC region. As this study had a broad geographic catchment area and took place during the first year of the COVID-19 pandemic, we were limited to collecting data online. This means that the perspectives of people without access to digital technologies may have been excluded from the findings. Collecting data online did, however, allow us to capture the perspectives of participants from all APEC member economies and from diverse backgrounds, providing a broad range of perspectives. Further research with participants facing barriers to the use of digital technologies is warranted to inform equitable access to DMH care.

Due to resource limitations, all data collection took place only in English, meaning that perspectives of non-English speakers from across the region were not captured. As part of the broader study, we collected similar data in Vietnam among Vietnamese speaking participants, which will be published elsewhere. Due to the constraints of the pandemic, we were unable to collect similar data in other partner countries. We believe this study nevertheless provides an important overview of regional priorities related to DMH equity.

Finally, the majority of participants represent people in HICs. Despite this, we achieved representation across the study data sets from all APEC member economies and presented disaggregated findings from Survey 2 to capture the perspectives of LMIC participants. The results demonstrated that priorities of LMICs respondents were similar to those of respondents in HICs. More research from LMICs related to equitable DMH delivery and access is needed, particularly as DMH options become more widely available in those settings.

## Conclusions

This study captures the perspectives of diverse stakeholders from across the APEC region, a large and diverse geographic area representing almost 40% of the world's population [74,82]. Findings of this study can inform both cross-regional and economy-specific initiatives to improve equitable access to DMH, and may be applied globally. While the APEC region is socioeconomically very diverse, study participants identified common challenges and suggest similar solutions related to the delivery of DMH care. This presents a substantial opportunity for collaboration and knowledge sharing across the region. The numerous challenges of the pandemic have called attention to the urgent need for mental health system strengthening and improved access to prevention, promotion, and care across the region. Despite substantial challenges, this represents is an unprecedented window of opportunity to advance mental health access across the region, including by embedding DMH within strengthened mental health systems. As DMH becomes more widely used and integrated into health systems, it is essential to ensure that equity is central to its design, delivery, evaluation, planning and policy. The prioritization of mental health and well-being by APEC represents considerable leadership potential to advance mental health system strengthening, including vis the use of DMH. The APEC region is therefore well-positioned to lead by example and must take a collaborative and proactive approach to advancing policy and practice to promote equitable access to DMH.

## Supporting information

**S1 Checklist. Inclusivity in global research questionnaire.**
(DOCX)

## Acknowledgments

The authors wish to acknowledge funding from the Canadian Institutes of Health Research and the Public Health Agency of Canada. We wish to thank all participants in this study who generously shared their experience and perspectives with us.

## Author Contributions

**Conceptualization:** Jill K. Murphy, Erin E. Michalak, Matias Irrarazaval, Mellissa Withers, Chee H. Ng, Andrew Greenshaw, John O'Neil, Vu Cong Nguyen, Harry Minas, Arun Ravindran, Jun Chen, Xing Wang, Tae-Yeon Hwang, Nurashikin Ibrahim, Simon Hatcher, Raymond W. Lam.

**Data curation:** Jill K. Murphy.

**Formal analysis:** Jill K. Murphy, Shirley Saker, Promit Ananyo Chakraborty, Tae-Yeon Hwang.

**Funding acquisition:** Jill K. Murphy, Erin E. Michalak, Matias Irrarazaval, Mellissa Withers, Chee H. Ng, Andrew Greenshaw, John O'Neil, Vu Cong Nguyen, Harry Minas, Arun Ravindran, Jun Chen, Nurashikin Ibrahim, Simon Hatcher, Raymond W. Lam.

**Investigation:** Jill K. Murphy, Raymond W. Lam.

**Methodology:** Jill K. Murphy, Erin E. Michalak, Raymond W. Lam.

**Project administration:** Amna Khan, Vanessa Evans.

**Supervision:** Jill K. Murphy, Raymond W. Lam.

**Writing – original draft:** Jill K. Murphy, Shirley Saker, Yuen Mei (Michelle) Chan.

**Writing – review & editing:** Erin E. Michalak, Matias Irrarazaval, Mellissa Withers, Chee H. Ng, Andrew Greenshaw, John O'Neil, Vu Cong Nguyen, Harry Minas, Arun Ravindran, Angela Paric, Jun Chen, Xing Wang, Tae-Yeon Hwang, Nurashikin Ibrahim, Simon Hatcher, Vanessa Evans, Raymond W. Lam.

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
