## [Decision Letter · Decision Letter 0]

30 Jan 2024

PGPH-D-23-02183

Advancing equitable access to digital mental health in the Asia-Pacific region in the context of the COVID-19 pandemic and beyond: A modified Delphi consensus study

Dear Dr. Murphy,

Thank you for submitting your manuscript to PLOS Global Public Health. After careful consideration, we feel that it has merit but does not fully meet PLOS Global Public Health’s publication criteria as it currently stands. Therefore, we invite you to submit a revised version of the manuscript that addresses the points raised during the review process.

The manuscript has been evaluated by one reviewer, and their comments are available below. The reviewer raised a number of concerns that need attention. They request additional information on methodological aspects of the study (such as information on the recruitment procedure and eligibility criteria) as well as revisions to the statistical analyses. 

Could you please revise the manuscript to carefully address the concerns raised?

Please note that we have only been able to secure a single reviewer to assess your manuscript. We are issuing a decision on your manuscript at this point to prevent further delays in the evaluation of your manuscript. Please be aware that the editor who handles your revised manuscript might find it necessary to invite additional reviewers to assess this work once the revised manuscript is submitted. However, we will aim to proceed on the basis of this single review if possible. 

We look forward to receiving your revised manuscript.

Kind regards,

Laura Kelly

Staff Editor

Journal Requirements:

1. Please provide additional details regarding participant consent. In the ethics statement in the Methods and online submission information, please ensure that you have specified what type of consent you obtained (for instance, written or verbal, and if verbal, how it was documented and witnessed). If your study included minors, state whether you obtained consent from parents or guardians. If the need for consent was waived by the ethics committee, please include this information.

2. Please include a complete copy of PLOS’ questionnaire on inclusivity in global research in your revised manuscript. Our policy for research in this area aims to improve transparency in the reporting of research performed outside of researchers’ own country or community. The policy applies to researchers who have travelled to a different country to conduct research, research with Indigenous populations or their lands, and research on cultural artefacts. The questionnaire can also be requested at the journal’s discretion for any other submissions, even if these conditions are not met.  Please find more information on the policy and a link to download a blank copy of the questionnaire here: https://journals.plos.org/globalpublichealth/s/best-practices-in-research-reporting. Please upload a completed version of your questionnaire as Supporting Information when you resubmit your manuscript.

3. Please send a completed 'Competing Interests' statement, including any COIs declared by your co-authors. If you have no competing interests to declare, please state "The authors have declared that no competing interests exist". Otherwise please declare all competing interests beginning with the statement "I have read the journal's policy and the authors of this manuscript have the following competing interests:"

4. Please amend your detailed Financial Disclosure statement. This is published with the article. It must therefore be completed in full sentences and contain the exact wording you wish to be published.

If you did not receive any funding for this study, please simply state: “The authors received no specific funding for this work.

5. We do not publish any copyright or trademark symbols that usually accompany proprietary names, eg (R), (C), or TM  (e.g. next to drug or reagent names). Please remove all instances of trademark/copyright symbols throughout the text, including © on page 44.

6. In the online submission form, you indicated that "Data are available upon request from the corresponding author". All PLOS journals now require all data underlying the findings described in their manuscript to be freely available to other researchers, either 1. In a public repository, 2. Within the manuscript itself, or 3. Uploaded as supplementary information.

Additional Editor Comments (if provided):

Reviewers' comments:

Reviewer's Responses to Questions

**Comments to the Author**

1. Does this manuscript meet PLOS Global Public Health’s publication criteria? Is the manuscript technically sound, and do the data support the conclusions? The manuscript must describe methodologically and ethically rigorous research with conclusions that are appropriately drawn based on the data presented.

Reviewer #1: Partly

2. Has the statistical analysis been performed appropriately and rigorously?

Reviewer #1: Yes

3. Have the authors made all data underlying the findings in their manuscript fully available (please refer to the Data Availability Statement at the start of the manuscript PDF file)?

Reviewer #1: Yes

4. Is the manuscript presented in an intelligible fashion and written in standard English?

Reviewer #1: Yes

5. Review Comments to the Author

Reviewer #1: This paper presents a modified Delphi study aimed at assessing needs and identifying priorities related to equitable digital mental health (DMH) access among marginalised populations in the Asia Pacific region during the first year of the COVID-19 pandemic. The paper is well written and structured, offering valuable insights into improving access to DMH. I commend the authors for their comprehensive approach and thoughtful analysis. However, I suggest the following for further clarification and enhancement of the manuscript:

1. Rationale for Using Delphi:

The authors stated, 'Our intention was not to reach full consensus but rather to generate a comprehensive picture of the landscape, needs, and priorities related to DMH access among at-risk groups across the region in the context of an unprecedented global health emergency.' However, given that reaching consensus is not the aim of the study, it would be beneficial to discuss whether the Delphi technique remains the most suitable method or if calling it a mixed-methods study would be more appropriate.

2. DMH Accessibility Across APEC Countries:

Could the author provide additional information regarding the accessibility of DMH services in each APEC country? Considering the significant contextual variations across countries in terms of technology ownership and DMH development, an overview of the general accessibility status in each country would enhance the contextual understanding of the study.

3. Recruitment Process:

The authors recruited a large sample size, which is impressive. However, the procedures to ensure the inclusion of the 'right individuals' (i.e., those with expertise regarding the topic, such as policymakers, service users, and those with lived experiences) were not clear. I suggest providing further information regarding the eligibility criteria for study participation and the screening process.

4. Data Analysis Software:

Please specify the software used for data analysis to ensure transparency and replicability.

5. Statistical Analysis:

I suggest the authors expand the statistical analysis to include between-group differences rather than solely comparing numbers. For example, ‘A majority indicated that they have noticed an increase, with no variation between LMIC (86.2%) and HIC (86.0%) respondents.’ (page 13, lines 327-328); ‘A majority (79.9%) responded that they had, with more people who had accessed DMH (91.2%) residing in LMICs compared with those in HICs (78.6%).’ (page 14, lines 333-334).

6. Survey 2 and Handling of Data:

Please explain the process of identifying and handling bot and spam survey responses in Survey 2, especially given the substantial number of such responses.

7. Reporting of Missing Data:

Are there missing data in the two surveys? If so, could the authors explain how missing data of the two surveys were handled?

In summary, while the study is commendable in its comprehensive approach to understanding DMH access in the Asia Pacific region during the pandemic, addressing these suggestions would enhance the clarity, transparency, and reliability of the research.

6. PLOS authors have the option to publish the peer review history of their article (what does this mean?). If published, this will include your full peer review and any attached files.

**Do you want your identity to be public for this peer review?** For information about this choice, including consent withdrawal, please see our Privacy Policy.

Reviewer #1: No

---

## [Decision Letter · Decision Letter 1]

3 May 2024

Advancing equitable access to digital mental health in the Asia-Pacific region in the context of the COVID-19 pandemic and beyond: A modified Delphi consensus study

PGPH-D-23-02183R1

Dear Dr. Murphy,

We are pleased to inform you that your manuscript 'Advancing equitable access to digital mental health in the Asia-Pacific region in the context of the COVID-19 pandemic and beyond: A modified Delphi consensus study' has been provisionally accepted for publication in PLOS Global Public Health.

Best regards,

Bibhav Acharya

Academic Editor

Reviewer Comments (if any, and for reference):

Reviewer's Responses to Questions

**Comments to the Author**

1. If the authors have adequately addressed your comments raised in a previous round of review and you feel that this manuscript is now acceptable for publication, you may indicate that here to bypass the “Comments to the Author” section, enter your conflict of interest statement in the “Confidential to Editor” section, and submit your "Accept" recommendation.

Reviewer #1: All comments have been addressed

2. Does this manuscript meet PLOS Global Public Health’s publication criteria? Is the manuscript technically sound, and do the data support the conclusions? The manuscript must describe methodologically and ethically rigorous research with conclusions that are appropriately drawn based on the data presented.

Reviewer #1: Yes

3. Has the statistical analysis been performed appropriately and rigorously?

Reviewer #1: Yes

4. Have the authors made all data underlying the findings in their manuscript fully available (please refer to the Data Availability Statement at the start of the manuscript PDF file)?

Reviewer #1: Yes

5. Is the manuscript presented in an intelligible fashion and written in standard English?

Reviewer #1: Yes

6. Review Comments to the Author

Reviewer #1: I appreciate the authors' responses and recommend that the paper be accepted.

7. PLOS authors have the option to publish the peer review history of their article (what does this mean?). If published, this will include your full peer review and any attached files.

**Do you want your identity to be public for this peer review?** For information about this choice, including consent withdrawal, please see our Privacy Policy.

Reviewer #1: No
